



# Constraining the wavefield of volcano-seismic events on Mt. Etna, Italy through rotational sensor and array observations

Nele Inken Käte Vesely[1], Eva Patrica Silke Eibl[1], Gilda Currenti[2], Mariangela Sciotto[2], Giuseppe Di Grazia[2], Matthias Ohrnberger[1], and Philippe Jousset[3]

[1]Institute of Geosciences, University of Potsdam, Potsdam, Germany
[2]Istituto Nazionale di Geofisica e Vulcanologia-Osservatorio Etneo, Catania, Italy
[3]GFZ Helmholtz Centre for Geosciences, Potsdam, Germany

**Correspondence:** Nele Inken Käte Vesely (vesely1@uni-potsdam.de)

**Abstract.**

Long-period (LP) events and tremor are characteristic seismic signals of active volcanoes, offering insight into underlying fluid-driven processes. Their emergent wavefield is complex and challenging to characterise. Seismic arrays as well as a rotational sensor with a co-located seismometer (6C station) can decipher LP event and tremor wave field composition. This study

aims to analyse and compare directional and phase velocity estimates by processing a 25-day long dataset from a rotational sensor and an array of seven broadband stations deployed at Mt. Etna, Italy, in August-September 2019. We derive the back azimuths (BAz) of LP events and tremor from both the seismometer array and the 6C station, and we compare these estimates with a reference BAz obtained from the network locations from the Istituto Nazionale di Geofisica e Vulcanologia-Osservatorio Etneo (INGV-OE) on Mt. Etna.

Volcanic tremor occurs in distinct phases with varying seismic and surface activity. Depending on the phase, either the array or 6C method provides reliable BAz estimates, agreeing well with the INGV-OE reference. We find that BAz estimates of both methods are shifted southward relative to the reference location for the LP events. We attribute the larger southward deviation observed in the 6C results to local heterogeneities which exert a stronger influence on the 6C station than on the array.

Based on the array derived slownesses we infer that the tremor and LP events mainly consist of surface waves. Further,

the rotational sensor recordings suggest a wavefield dominated by SH-type waves. In combination with the observed temporal evolution of the 6C phase velocity in narrow frequency bands, we infer Love-wave dominance. This study highlights the value of a rotational sensor to constrain the wavefield in a deterministic way in a complex volcanic environment.

## 1 Introduction

Volcanic tremor often marks the build-up to, and accompanies eruptive activity (McNutt, 2002; Zuccarello et al., 2022). At

many volcanos, it is closely correlated with explosive phenomena such as paroxysms, and lava fountaining (Zobin, 2017). Tremor is characterized as an emergent, continuous signal lasting minutes to days, typically with peak power in the 1-5 Hz range, though this varies between volcanic systems (McNutt, 2002). Another class of typically observed volcano-seismic signal are long-period (LP) events which exhibit dominant frequencies from 0.5 to 5 Hz (Chouet and Matoza, 2013). Unlike tremor,





LP events can sometimes show a clear onset of the p-wave arrival (Chouet, 2009). Different location approaches to handle LP
events and volcanic tremor with seismometer networks exist (e.g. Battaglia et al. (2003); Di Grazia et al. (2006); De Barros
et al. (2009); Cannata et al. (2010)). The wavefield of those event types is complex and lacking observations at appropriately
sized arrays or 6C recordings, so that the detailed composition of wave types often remains unclear.

Conventional seismometer networks record unrest on volcanoes, but for detecting weak volcano-seismic signals, smaller,
denser aperture networks are more promising (Chouet, 1996b; Saccorotti et al., 2001; Wassermann, 1997). Arrays, originally
set up in the 1960s to detect nuclear tests (Douglas, 2002), have since become key tools for studying the Earth's structure and
exploring volcanic activity (Rost and Thomas, 2002). These seismic arrays consist of multiple seismometers placed at small
station spacings, ranging from a few tens to hundreds meters. An incoming wave is recorded coherently on all seismometers
within a frequency range that depends on the station spacing and the seismic ground velocities (McNutt, 2002). In such a set-up
the ground motion along three orthogonal axes is measured. Weak or emergent signals can then be detected more easily by
stacking data. Information about the wavefield can be gathered, migrating sources tracked and source directions or incidence
angles determined (McNutt, 2002; Rost and Thomas, 2002). Seismic arrays are advantageous for exploring volcanic tremor
(Chouet, 1988, 1996a; Ferrazzini et al., 1991).

Arrays have been used to locate (Chouet, 2003; Métaxian et al., 2002) and distinguish tremor wave-types (Saccorotti et al.,
2004), track migrating tremor sources (Eibl et al., 2017b, 2023, 2020; Zuccarello et al., 2022) and differentiate between dif-
ferent simultaneously recorded tremor (Eibl et al., 2017a) or between tremor and long-period event sources (Almendros et al.,
2014). Long-period seismicity can likewise be analysed by arrays (Almendros et al., 2001; Chouet, 2003). Even though arrays
enable a better understanding of the wavefield and volcanic structures, their installation and maintenance is challenging and
costly in volcanic settings which are mostly difficult to access or do not permit specific array geometries. The performance
depends on the operation of all seismometer stations. In case of station failure, the array geometry, aperture and thus sensitivity
to a certain frequency range changes (Wassermann et al., 2016).

If space is limited or the site is difficult to access, point measurements using rotational sensors might offer an alternative to
seismic arrays (Cochard et al., 2006). In recent years, portable rotational sensors have become available allowing the recording
of the full rotational wave field with high resolution around three orthogonal directions. These fibre optic gyroscopes can be co-
located with a seismometer to measure six degrees of freedom: three translations and three rotations. A six component station
requires less installation effort and maintenance in comparison with a multiple station array, while being able to separate the
wavefield and calculate directions of arriving signals. Processing techniques with six component measurements (Hadziioannou
et al., 2012; Wassermann et al., 2020) aim at providing the same information as normally obtained by a multi-station array
(Brotzer et al., 2025; Cochard et al., 2006). Various laboratory (Bernauer et al., 2018) and passive and active field experiments
(Izgi et al., 2021, 2025) with rotational sensors have confirmed low self-noise, a broad frequency range and realistic event
direction estimation. First experiments on volcanoes have been carried out on Kilauea (Wassermann et al., 2020), Stromboli
(Wassermann et al., 2022) and Etna (Eibl et al., 2022a) which have shown great advances in wavefield analysis using a rotational
sensor with a co-located seismometer. Wassermann et al. (2022) were able to detect gas jetting events not recorded on other

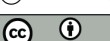



instruments, leading to the build-up of a new vent on Stromboli volcano. By using a rotational sensor, Eibl et al. (2022a) separated the wavefield of LP events and tremor at Mt. Etna which is helpful in characterizing the source system.

A recent study by Keil et al. (2022) compares rotational sensor and array recordings of anthropogenic noise. Resulting P- and S-wave velocity profiles from arrays and the rotational sensor and co-located seismometer (6C) station were in agreement, thus confirming the suitability of using the six component station approach (Keil et al., 2022). The self-noise of the rotational sensor was a major constraint for frequencies below 5 Hz (Keil et al., 2022). Bernauer et al. (2021) identified a limitation of 4 Hz in the rotational sensor's response during their comparative study of various sensors used for measuring rotational motion.

Recordings of rotation and rotation rate are furthermore stronger affected by local heterogeneities than displacement, velocity and acceleration recordings (Singh et al., 2019). Another study by Brotzer et al. (2025) compared a rotational sensor and seismic array with respect to Signal-to-Noise Ratio (SNR) and back azimuth calculation of a local and a regional earthquake. Stations with downtime had an effect on the array derived rotation (ADR) (Brotzer et al., 2025). The accuracy of ADR furthermore depends on the array aperture, distance to and type of analysed signals (Suryanto et al., 2006). In contrast, measurements

obtained from a single rotational sensor are independent of inter-station geometry or availability, as they directly record ground rotations at a single point.

    In volcanic environments, source signals might be stronger and additional information about source processes could help improve eruption forecasting. We investigate the performance of a 6C station in comparison to an array by calculating back azimuths and phase velocities of LP event and tremor signals. The aim is to test for a first time in a volcanic environment,

whether the 6C approach yields reliable back azimuth and velocities compared with a conventional seismic array and whether they furthermore match with the reference locations from the Istituto Nazionale di Geofisica e Vulcanologia-Osservatorio Etneo (INGV-OE)'s network. We briefly introduce the area of our study at Etna volcano, summarize the INGV-OE's network processing for localizing the long-period events and volcanic tremor and explain the rotational sensor and the array processing in Section 2. Results of the back azimuth comparisons to the INGV-OE locations are presented in Section 3 for tremor and

LP events. An interpretation and discussion is given in Section 4 along with an explanation of limitations in this study. We conclude that a rotational sensor in combination with a seismic array can be used to pin down the wave types in a complex wavefield (5).

## 2 Methodology

### 2.1 Seismic installation on Etna volcano

Mount Etna is one of Europe's most active and visited volcanoes (Scarfì et al., 2023). This stratovolcano is located on the Italian island of Sicily (Fig. 1a) and is characterized by Strombolian activity, lava flows and fountains, as well as earthquakes from the sliding eastern flank. Touristic areas on the volcano's flanks (Currenti and Bonaccorso, 2019), nearby villages and the town of Catania are at risk under the volcanic activity. Thus, understanding volcanic activity is undoubtedly important for mitigating hazard. Although fissure eruptions occur, the most frequent eruptive activity is concentrated in the main summit



area within Bocca Nuova (BN), North East crater (NEC), Voragine (VOR), South East crater (SEC) and the New South East crater (NSEC), as illustrated in Fig. 1e.

From August to September 2019, a dense broadband array and a rotational sensor were installed in Piano delle Concazze on Etna (Currenti et al., 2021; Eibl et al., 2022a). Six 3C Trillium Compact 120 s seismometer (Nanometrics™) with DiGOS DATA-CUBE3 digitisers are used as a small aperture array where one of them is co-located with the rotational sensor (Fig. 1d).

The inter-station distance is 60-90 m and the seven seismometers are buried at 40 cm depth in pyroclastic deposit (Currenti et al., 2021). Their maximum elevation difference is 10 m. One 3C blueSeis-3A rotational sensor *RS1* (formerly: iXblue, now: exail) was co-located with the 3C Trillium Compact 120s seismometer named *Bb17* on the same quadratic, granitic plate and recording with a 200 Hz sample rate (Eibl et al., 2022a). We refer to the rotational sensor and the co-located seismometer as the six component (6C) station (Fig. 1c). In the recording time period, an eruption started on 4 September 2019 with weak ash

emission at the NEC and Voragine crater. Around two days later it was accompanied by Strombolian activity at the NEC and finally led to lava effusion from 18 September 2019 until around 21 September 2019 at Voragine crater (Eibl et al., 2022a). Eibl et al. (2022a) have shown using a rotational sensor, that LP events at Mt. Etna mainly have a strongly polarized wavefield in the horizontal plane and are therefore composed of SH-type waves. The tremor wavefield is composed of SH-type waves in August and early September and features a mixed wavefield from 9 September 2019 until 13 September 2019 (Eibl et al.,

2022a).

## 2.2 Root mean square calculation and Signal-to-Noise ratio

The root mean square (RMS) is a statistical measure to quantify the average signal ($S$) amplitude. The RMS value reflects the signal energy over a specific time period. It is calculated by squaring each signal value $i$, averaging these squares over all

samples $I$, and then taking the square root:

$$RMS_{signal} = \sqrt{\frac{1}{I}\sum_{i=0}^{I} S_i^2}. \tag{1}$$

The RMS of the long duration tremor is calculated with sliding windows of 30 minutes and 20% overlap. We plot this for visualisation.

For each LP event we calculate the RMS in one 5 second long time window containing the signal and the RMS in a 5 second

long time window with noise that starts 25 second prior to the start of the signal window. Finally, we use these to calculate the Signal-to-Noise ratio (SNR) by:

$$SNR_c = \frac{RMS_{signal}}{RMS_{noise}}. \tag{2}$$

The SNR indicates how well a signal is visible in comparison with the noise level. For all LP events the SNR is calculated for each component ($c$). LP event data is tapered, instrument corrected for the seismometers only, then filtered by a two-pole





**Figure 1.** Overview of the network set-up. (a) Etna volcano on Sicily, Italy. (b) Locations of 16 stations of the INGV-OE network around the main craters used for tremor processing (little white squares). Stations within the white frame box were used for the catalogue LP event detection. The satellite image is taken from © Google Earth 2019. (c) Set-up of the 6C station. (d) Location of the array stations (blue dots), the 6C station (red triangle) and faults (red and dotted yellow lines). (e) Summit crater locations.





zero-phase Butterworth bandpass filter from 0.6 to 10 Hz for the array and from 0.6 to 1.4 Hz for the rotational sensor. For the tremor data the same steps are applied, but the data is filtered from 0.5 to 10 Hz for the seismometer and from 0.5 to 2.5 Hz for the rotational sensor. The tremor data is furthermore resampled from 200 Hz to 50 Hz.

## 2.3  INGV-OE network processing

First, the amplitude-based tremor localization method is described, followed by the INGV-OE network's use of radial sem-
blance for LP event localization. The volcanic tremor source is routinely located using a subset of 16 permanent seismic network stations equipped with Trillium seismometers (Nanometrics™) operated by INGV-OE and installed at altitudes from 1200 m to 3000 m altitude in a range of 10 km from the summit craters (Fig. 1b). The location procedure is based on the seismic amplitude decay method and implemented as a 3D grid search (Battaglia et al., 2005; Di Grazia et al., 2006; Cannata et al., 2013). The spatial grid is centred at the summit crater Voragine and extends horizontally 6x6 km and 3 km vertically with
a node spacing of 250 m. Amplitudes are measured from the vertical component data of 30-min seismic signal windows. The data are Butterworth filtered (0.5-2.5 Hz) and used to fit the RMS amplitude to source-station distance. This step is performed for each working station and each node of the grid, that is iteratively assumed to be the source of the volcanic tremor. The goodness of the linear regression fit is represented by the value of $R^2$, and the source is identified as the centroid of the grid nodes whose $R^2$ values do not differ more than 1% from the maximum $R^2$ value (Di Grazia et al., 2006; Cannata et al., 2013).
To obtain reliable solutions, thresholds of $R^2$ goodness $\geq 0.9$ and a minimum station number of 13 are set. An uncertainty of 500 m, based on Jackknife estimation (Efron, 1982), is assumed to assess tremor location stability. The event location uncertainty is on the order of a few hundred meters in latitude and longitude, but can reach up to approximately 1 km in depth (Di Grazia et al., 2006; Cannata et al., 2018; Cannavo' et al., 2019).

For the location of the LP events, we consider a catalogue of amplitude transients with peak frequency in the spectral
band 0.2-1.1 Hz detected and extracted from the continuous signal by means of the Short Time Average/ Long Time Average (STA/LTA) energy algorithm (e.g.: Trnkoczy 2012). The event source location method is based on the similarity among waveforms at the different stations on the particle motion of very long-period (VLP) events (Cannata et al., 2009). It is a grid search method (with the same grid used for volcanic tremor location) and implements, using the three components of the filtered signal, the radial semblance function (Cannata et al., 2013). The source is placed in the node of the grid characterized by the
maximum semblance value. For this procedure, only summit stations are used, which in the time interval of the field campaign were only four (see the white frame box around the summit in Fig. 1b). To obtain robust locations, solutions with a radial semblance higher than 0.6 are considered reliable and locations outside the summit crater area are excluded. 371 LP events were detected and located during the field experiment. An uncertainty of 500 m for the LP event locations is included in the analysis. We associate the lateral resolution to the radius of the first Fresnel volume calculated with an assumed velocity of
1000 m/s (Eibl et al., 2022a), 1 Hz frequency and 500 m source depth resulting in 500 m uncertainty (Yilmaz and Doherty, 2001):





$$r = \frac{v}{2} \cdot \sqrt{\frac{2 \cdot z}{f \cdot v}}, \tag{3}$$

with $r$ as the point distance, in this case the uncertainty, the velocity $v$, source depth $z$, and frequency $f$ (Yilmaz and Doherty, 2001). Further parameter combinations for this calculation can be found in Suppl. Table S1. The average of all combinations

is 480 m.

The back azimuth (BAz) of the incoming wavefront is defined as the angle between North and the direction to the epicentre from the station (Rost and Thomas, 2002). We calculate the INGV-OE back azimuth for LP events and tremor as the direction measured clockwise from North to the INGV-OE locations from the array centre point of view. This enables direction comparison with the array and 6C BAz estimates, as the back azimuth to the array and the rotational sensor respectively only differ by

$0.02°$ to $0.05°$ around the main crater region.

## 2.4 Rotational sensor processing

SH-type waves are recorded on the vertical component of the rotational sensor and on the horizontal components of the seismometer. We therefore correlate the vertical rotation rate with the transverse acceleration for a given angle (Cochard et al., 2006; Hadziioannou et al., 2012; Izgi et al., 2021; Keil et al., 2022) and save the correlation coefficient. This procedure is

repeated for all angles from 0 to 360 degree. Finally, the angle that yields the highest correlation between transverse acceleration and vertical rotation rate is assumed as the source back azimuth (Hadziioannou et al., 2012).

By dividing the transverse acceleration $\ddot{u}_T$ by the vertical rotation rate $\dot{\omega}_Z$, assuming a transversely polarized plane wave with amplitude $A$, wavenumber $k$ and phase velocity $c$ propagating in x-direction, we obtain Eq. 4, 5. The ratio of $\ddot{u}_T$ and $\dot{\omega}_Z$ is constant and equal to $-2c$ (Eq. 6).

$$\ddot{u}_T = -k^2 c^2 A sin(kx - kct) \tag{4}$$

$$\dot{\omega}_Z = \frac{1}{2} k^2 c A sin(kx - kct) \tag{5}$$

$$\frac{\ddot{u}_T}{\dot{\omega}_Z} = -2c \tag{6}$$

The phase velocity is calculated from the RMS amplitude ratio of rotation rate and transverse acceleration motions for the angle that yields the highest correlation of the waveforms (Hadziioannou et al., 2012). We refer to this method as the 6C approach.

First, we demean and detrend the data linearly and apply a cosine taper. We then use a bandpass zero phase filter with two corners. We calculate the BAz in steps of $1°$ and with a correlation coefficient threshold of 0.15. Additionally, we calculate



the standard deviation of all BAz values within each sliding time window if their correlation values are at least 95% of the maximum correlation of that time window.

Rotational sensor back azimuths for the tremor were derived in a frequency range of 0.5 to 2.5 Hz with sliding time windows of 30 minutes. We tested different LP event frequency ranges for the 6C processing, which resulted in best BAz results for the 0.6 to 1.4 Hz range. We also compared to an INGV-OE catalogue of the same frequency range, which did not enhance the comparison, thus adhere to the initial 0.2-1.1 Hz INGV-OE catalogue. Sliding window length in this case relates to the minimum frequency of interest. We choose a window length of three times the corresponding signal period at 0.6 Hz, thus 5 s with 90% overlap. For every LP event we obtain one back azimuth value and one phase velocity value by selecting the time of highest correlation between the vertical component of the rotational sensor and the rotated horizontal (transverse) components of the co-located seismometer bb17 within the manually selected event time by the Pyrocko seismogram workbench (Heimann et al., 2017).

## 2.5   Array analysis

Plane wave parameters are estimated from array data by a standard band-limited slowness power spectrum analysis in frequency domain. For this purpose we maximize the beam power BP, computed as:

$$BP(\boldsymbol{s}) = \sum_{m=m1}^{m2} \left| \sum_{n=1}^{N} X_n(m\Delta f) \exp(-j2\pi m\Delta f \boldsymbol{r}_n \boldsymbol{s}) \right|^2 \tag{7}$$

for slowness vectors $\boldsymbol{s} = (s_x, s_y)^T$ in a slowness grid in the horizontal plane of wave propagation. In Eq. 7 $X_n(m\Delta f)$ represents the Fourier coefficient for the observed signal at station $n$ at position $\boldsymbol{r}_n$ and discrete frequency $m\Delta f$. The shifted signals are summed over the total number of stations $N$ within the array. The squared sum of the stacked signals is then further summed over the frequency band from $m1$ to $m2$ corresponding to the indices for the lowest and highest frequencies of interest. $\Delta f = 1/T$ represents the frequency resolution of the Fourier transformed signal of length $T$.

The slowness vector corresponding to the maximum power in the map points to the propagation direction of the best fitting plane wave and its absolute value indicates the inverse of its apparent velocity (Wassermann, 1997). The horizontal slowness and back azimuth from azimuth $\Psi$ can be obtained from the components of the slowness vector (Beyreuther et al., 2010; Chouet et al., 1997) as:

$$s_{hor} = \sqrt{s_x^2 + s_y^2}. \tag{8}$$

and

$$\Psi = \frac{180°}{\pi} \cdot arctan(s_x, s_y). \tag{9}$$

For details we refer to Rost and Thomas (2002), Schweitzer et al. (2012) and Buttkus (2000).



We perform array processing from 24 August 2019 to 14 September 2019 on tremor signal with the vertical component seismometer data. Unlike the horizontal components, the vertical components are not expected to be strongly influenced by topography, tilt (Wassermann, 1997) or misalignment with geographical north. The slowness grid boundaries are defined to $\pm 3\,\text{s\,km}^{-1}$ with a step size of $0.01\,\text{s\,km}^{-1}$. A frequency range from 0.5 to 10 Hz was selected, as the tremor signal is strongest from 1 to 5 Hz (Suppl. Fig. S1), but the array response shows best directional sensitivity with low power side lobes from 0.6 to 10 Hz (Suppl. Fig. S2). We furthermore want to distinguish between noise and signal and chose a broader frequency range. We process 30-min sliding windows that have no overlap to compare with the INGV-OE results as in Eibl et al. (2022a). We downsample the tremor data from 200 Hz to 50 Hz after applying a lowpass filter of 20 Hz to avoid aliasing. Uncertainties within the array processing are calculated from back azimuth and slowness values for each grid point if the power is at least 95% of the maximum value (Eibl et al., 2017b).

For analysing 371 LP events the same slowness grid was used. A sliding time window of 5 seconds and 90% overlap, as suggested in Eibl et al. (2022a), is applied. Results are displayed in the centre of the sliding window of 5 s which shifts by 0.5 s during each iteration. Although the highest LP event amplitude is expected from 0.6 to 1.4 Hz by the spectrogram and power spectral density (PSD) analysis (Suppl. Figs. S3, S4, S5), we apply a maximum frequency of 10 Hz for the array processing according to the aperture of the array (Suppl. Fig. S2) and a minimum frequency of 0.6 Hz due to poor resolution at lower frequencies similar to Zuccarello et al. (2016). The event signal's frequency content is expected to dominate the recording, while noise before and after spans a broader range. Therefore, we can compare with the 6C results. For comparison with the rotational sensor and the routine monitoring locations by INGV-OE, we take the back azimuths during the LP event based on the manually selected picks in the Pyrocko seismogram workbench (Heimann et al., 2017). For every LP event, the BAz and slowness value is taken from the time of maximum relative power (semblance) within the time window of the event. As a result, we obtain one back azimuth and slowness value for each event.

## 3 Results

### 3.1 Volcanic tremor analysis

#### 3.1.1 Volcanic activity, tremor amplitude and location

We use the observed volcanic activity described by Eibl et al. (2022a) to define eruption phases (Table 1). We then attribute the tremor locations (Fig.2) derived by the INGV-OE to these phases. Prior to the first observed activity, during phase 0 tremor was located around NSEC, SEC and reaching towards VOR crater (white circles in Fig. 2). From 4 September 2019, weak ash emission was noted. Until 6 September 2019 locations during this phase 1 activity cluster around SEC and mostly the NSEC. During phase 2 (yellow circles in Fig. 2), the INGV-OE detected Strombolian activity and tremor east of NSEC, with some locations further north. Strong Strombolian activity was observed from 9 September until 12 September (orange circles in Fig. 2). Locations during these days spread between an area from NEC to north of NSEC and in between, but mostly accumulate below the NEC. This phase was followed by Strombolian activity until 13 September at the NEC, then at VOR





**Figure 2.** Illustration of the tremor locations (white (Phase 0), gray (Phase 1), yellow (Phase 2,4) and orange (Phase 3) according to eruption phases), array stations (blue dots), rotational sensor (red triangle), array centre (yellow dot). Uncertainties of 500 m are drawn around the locations as red circles. Summit craters are indicated.

crater. Tremor locations for this last phase comprise a broad area between the NEC, VOR and SEC with some locations west of NEC and are displayed as yellow circles with black outline.

Both instruments' components, the vertical rotation (HJZ) and the vertical velocity (HHZ) show similar RMS amplitude changes over time (Fig. 3a). RMS data for all components of Bb17 and RS1 show similar trends (Suppl. Fig. S7). The amplitude is low until 3 September 2019 3 PM UTC. Then the amplitude rises and ash emissions (phase 1 as gray colour in Fig. 3a) start on 4 September at NE crater. At the first tremor increase, the frequency ranges from 1-2 Hz and is strongest on the horizontal array components and the vertical rotational sensor component. Phase 2 with Strombolian activity starts on 6 September, but the tremor RMS does not increase further compared to phase 1. A minor peak in amplitude is visible on 8 September from 02:58 to 06:30 UTC. A significant increase on 9 September at 01:30 UTC marks the beginning of phase 3 with sustained



| phase | timing | eruption state | visible activity | tremor locations |
|-------|--------|----------------|------------------|------------------|
| phase 0 | before 2019-09-04 | pre-eruption | no visible activity | between NSEC, SEC, VOR |
| phase 1 | 2019-09-04 to 2019-09-06 | eruption start | ash emission (NEC) | NSEC to SEC |
| phase 2 | 2019-09-06 to 2019-09-09 | eruption evolving | Strombolian activity (NEC, VOR) | South, East of NSEC |
| phase 3 | 2019-09-09 to 2019-09-12 | eruption climax | strong Strombolian activity (NEC) | mostly NEC |
| phase 4 | 2019-09-12 to 2019-09-14 | stable eruption, declining | Strombolian activity (NEC,VOR) | South and West of NEC |

**Table 1.** Eruption phases, classifications of superficial activity and BAz based on the INGV-OE locations.

Strombolian activity at the North East crater until 10 September shown in orange. Strong amplitude tremor energy reaches
from 1 Hz to 4 Hz (Suppl. Fig. S1 and S7). Large amplitudes are visible until 12 September, when another minor peak appears, after which the amplitude decreases to a level similar to the eruption start. This is assigned to phase 4 until the end of analysed time on 14 September 2019.

### 3.1.2 Tremor back azimuths

We compare the time series of tremor back azimuths derived from (i) array processing, (ii) a rotational sensor and (iii) the
255 INGV-OE network catalogue locations from the end of August 2019 to 14 September 2019 (Fig. 3).

From phase 0 to phase 1 we observe back azimuth changes from 210° decreasing to 190° on 4 September (Fig. 3b). In phase 1 and 2 the BAz values vary between 190° and 200°. On 8 September during the RMS peak from 3 AM to 6 AM UTC, BAz values increase from 190° to 225°, pointing to the northern (NE, BN, VOR) craters. Afterwards the BAz values migrate again to around 190° until the next day, when the BAz values then continuously move from 200° to 230°, the directions of the northern
craters, coinciding with the increase of Strombolian activity observed at those. From 9 September (phase 3) the BAz values are stable at around 230° to 235°, pointing still to the northern craters. On 12 September (phase 4), back azimuth values start to decrease from around 240° to 210° until 14 September. These INGV-OE reference back azimuths and uncertainty ranges are shown in Fig. 3c, d.

The array-derived BAz (Fig. 3c) fit the INGV-OE reference range until the beginning of phase 1. From 4 until 8 September,
BAz with higher relative power than during phase 0 show directions towards the southern craters and are within the upper limit of the INGV-OE reference. On 8 September with the first strong tremor amplitude peak, BAz and INGV-OE values overlap. Directional estimates continue to agree in phase 3 in which Strombolian activity intensified. Nevertheless, BAz scatter more strongly between the reference range from 210° to 240°. In phase 4 Strombolian activity was still noted, but switched from NEC to VOR crater between 12 and 13 September, represented by decreasing BAz values from 230°-240° to 210°-220°.

During phase 0, phase 1 and phase 2 6C-derived BAz (Fig. 3d) vary mostly between 185° and 200° within the INGV-OE reference range. Correlation between the horizontal seismometer and vertical rotational sensor components is highest during







**Figure 3.** Back azimuth tremor comparison from 31 August 2019 until 14 September 2019. (a) RMS of seismic data of the Bb17 seismometer (HHZ component, blue) filtered from 0.5 to 10 Hz and rotational sensor (HJZ component, black) filtered from 0.5 to 2.5 Hz and observed activity as coloured blocks. (b) Black points show the INGV-OE BAz, with the respective uncertainty bars of 500 m. Coloured horizontal lines indicate the BAz of the craters. (c) Back azimuths derived using the vertical array component, coloured according to the semblance with gray vertical bars indicating the associated uncertainty. Gray lines in the background show the INGV-OE BAz uncertainty range. (f) The back azimuth values derived using the 6C-method, coloured according to correlation between the transverse acceleration and the vertical rotation rate.





these times. From phase 3 to phase 4, the 6C BAz are lower than the INGV-OE range. Directions point to NSEC and SEC instead of the northern craters where strong Strombolian activity was observed.

In summary, phase 0 features different direction trends for all methods. During phase 1, 6C-derived BAz fit the reference better than the array results, even though the tremor onset on 3 September is clearly visible on both instruments (Suppl. Figs. S1, S6, S7). Phase 2 is best displayed by the 6C BAz as well, but the first RMS amplitude peak on 8 September and corresponding direction changes are better represented by the array-derived BAz. In phase 3 and 4 with highest tremor amplitude and following decreasing amplitudes we see closest agreement between the array-derived BAz values and the INGV-OE locations' BAz. For all methods, the semblance and maximum correlation are highest from 4 until 8 September, phases 1 and 2. However, some time windows during phase 3 also show high semblance for the array.

### 3.1.3 Tremor slownesses and phase velocities

We compare slowness and phase velocity results with the visible activity and the RMS tremor amplitude (Fig. 4). The hypocentral distance (Fig. 4b) of tremor INGV-OE locations to the array centre varies from 2.1 km to 2.5 km until the end of phase 2, when it decreases with the first RMS amplitude peak to 1.6 km distance before increasing to 2 km again. At the beginning and end of phase 3, the distance drops from ≈2 km to ≈1.6 km each time. At the onset of phase 4, tremor is located at a distance between ≈2.2 km and ≈2 km. Uncertainties of the distance are of 500 m. The vertical resolution is constrained by 1000 m uncertainty. Accordingly, small-scale variations in the estimated source position fall within the range of methodological uncertainty and cannot be considered resolvable with confidence. The strongest change of the distance results from the deepening of the tremor source during phase 3 with sustained Strombolian activity. The tremor migrated from the surface at ≈2.9 km down to 1.6 km altitude, a trend which is despite the uncertainties, reliable.

Array-derived slowness values (Fig. 4c) decrease from 1.6 s km$^{-1}$ to 1.0 s km$^{-1}$ during phase 0 and phase 1. With exception of the RMS peak, slowness values are stable mostly at 1.2 s km$^{-1}$ during phase 2. Highest slowness values of around ≈1.7 s km$^{-1}$ are reached during the RMS peak in phase 2 and with the strong tremor amplitude increase at the beginning of phase 3. During phase 3 slowness values vary only between 1.4 s km$^{-1}$ and 1.2 s km$^{-1}$. With the beginning of phase 4, slowness values increase linearly from 1.2 s km$^{-1}$ to 1.6 s km$^{-1}$. Interestingly, slowness values are stable at the times of highest relative power, which coincides with the beginning of visible activity and the strong Strombolian activity.

Phase velocities obtained from the 6C method (Fig. 4d) vary mostly between 0.35 km s$^{-1}$ and 0.4 km s$^{-1}$ in phase 0, 1 and 2. With the beginning of phase 3 the velocities increase from 0.35 km s$^{-1}$ to 0.45 km s$^{-1}$. The shape of velocity changes seem to represent the RMS tremor amplitude changes with three peaks in phase 3. In phase 4 the values drop linearly from 0.4 km s$^{-1}$ to ≈0.33 km s$^{-1}$.

The most pronounced changes in tremor distance, altitude, array slowness, and 6C phase velocities occur during the end of phase 2 and throughout phase 3, coinciding with elevated RMS tremor amplitude and intense Strombolian activity. Velocities derived from the combination of the transverse acceleration and vertical rotation correlations (6C velocities) correlate with peaks in tremor amplitude.



**Figure 4.** Slowness and phase velocity comparison for the tremor from 31 August until 14 September 2019. (a) RMS of seismic data of the Bb17 seismometer (HHZ component, blue) with a 0.5 to 10 Hz and rotational sensor (HJZ component, black) with a 0.5 to 2.5 Hz filter and observed activity as coloured blocks, indicated as vertical dashed lines in the following subplots. (b) Hypocentral distance from the array to the tremor locations coloured by event altitude. (c) Slowness values derived using the vertical array component, coloured by semblance. (d) The phase velocities derived using the 6C-method, coloured by correlation between the transverse acceleration and the vertical rotation rate.





## 3.2 Long-period event analysis

In the following we first discuss the LP event signal-to-noise ratios, and show an exemplary long-period event to explain the processing used to obtain one event back azimuth and phase velocity value. We then present an comparative overview of all the 371 LP events analysed.

### 3.2.1 Signal-to-noise ratio of LP events

While most events show comparable SNR across components, some exhibit highest SNR on HHZ, supporting the use of the vertical component for the array analysis (Suppl. Fig. S8). The SNR from the HJZ component of the rotational sensor mostly exceeds those of the HJN and HJE components by two. The SNR for the selected LP event on 2 September 15:51:10 UTC is shown in Suppl. Fig. S9, S10 and S11. For that LP event, SNR values are highest for the array's HHN and HHZ components and for the rotational sensor's HJZ and HJN components. This event is detected on all components, but exhibits highest energy on the horizontal components of the co-located seismometer and on the vertical component of the rotational sensor as shown in the spectrogram (Suppl. Fig. S3).

### 3.2.2 Exemplary LP event back azimuths and correlations

The HHZ component of one array station is shown alongside the vertical component of the rotational sensor (Fig. 5a, b). High amplitudes indicate the INGV-OE picked LP event within the grey dashed lines. Another lower amplitude event is also visible approximately 10 seconds earlier on both instruments' components. Array-derived back azimuths fluctuate between mostly 180° to 240° (Fig. 5c). At the event start, the BAz fits well with the INGV-OE reference (red line) and semblance between the array stations is high. The small time shifts between the high semblance time windows and the event waveforms result from the sliding windows (Fig. 5e) and 0.1 overlap applied in the analysis. A slight increase in semblance is also observed approximately 10 seconds prior to the event, coinciding with the potential recording of another LP event. Slowness values are lowest at the event start with $\approx 0.5$ s km$^{-1}$, increasing linearly to 2 s m$^{-1}$ towards the event's end. No such slowness change is noted 10 seconds prior the picked event. The rotational sensor back azimuths (Fig. 5d) range is broader than the array results with $\approx 150°$ to 230°. At the event start 6C BAz indicate a direction of 201°, thus pointing 24° further south than the INGV-OE reference. Phase velocities are highest at the event start reaching up to $\approx 0.8$ km s$^{-1}$ and 10 seconds prior at 0.6 km s$^{-1}$.

### 3.2.3 LP event catalogue: temporal and spatial distribution

We analysed a total of 371 LP events from 24 August 2019 to 14 September 2019, with no LP events in the INGV-OE catalogue from 5 to 14 September, during the eruption phase (Fig. 6a). Additional LP events are visible during this phase but not listed in the catalogue, either because they were not possible to locate with low amplitude in comparison to the tremor amplitude (Sciotto et al., 2022) or because of the catalogue frequency range set for this study. Other small-amplitude signals are visible, but have not been detected by the network, see Fig. 5a, b at 15:51:05. We do see similar small-amplitude signals for $\approx 103$ of the 371 LP events picked. All events have been located by the INGV-OE routine at eleven locations (Fig. 6b) at altitudes varying





**Figure 5.** Comparison of the long-period event at 15:51:10 on 2 September 2019. (a) Bandpass-filtered seismograms (2nd-order) are shown for the vertical component of seismometer Bn17 (0.6 – 10 Hz) and (b) for the vertical component of the rotational sensor (0.6 – 1.4 Hz). Gray, dashed lines indicate the event time windows, the black line (c) the sliding window. (c, d) BAz and slowness (phase velocities) derived using the vertical components of the array (c) and the 6C method (d). Colours indicate the semblance for the array and maximum correlation for the 6C results. The BAz of highest power (correlation) is annotated. The INGV-OE value is shown as a red line with its uncertainty as a gray rectangle.



from 2.75 km to 3.25 km, but mostly at 3 km. 189 LP catalogue events occurred from 27 until 29 August (Fig. 6a). Prior to the first visible activity on 4 September, LP BAz from the INGV-OE catalogue point to the north-east, Bocca Nuova and Voragine craters alternately. The first LP events further south appear from the evening on 3 September until early 4 September, however other LP events are still detected at the northern craters. Only four LP events are located by the INGV-OE at the southern craters in total (Fig. 6b).

### 3.2.4 Back azimuths of the LP events

If the back azimuth estimated from the array or 6C method for an LP event falls within the corresponding uncertainty range of the INGV-OE BAz, the event is counted as a valid result for the respective method, as indicated in the legend. For all LP events the uncertainty range of INGV-OE BAz varied from ±10-14°. The respective BAz are obtained at the event time of highest semblance (array) or highest correlation (6C). Of 371 LP events, we obtained 119 valid Baz from the HHZ components and 67 valid BAz from the 6C method, see legends in Fig. 6c, d. Craters are indicated as coloured horizontal lines for orientation. INGV-OE locations (black points) point to the northern craters until 4 September when few LP events are located at SEC (Fig. 6c, d). HHZ components' BAz point mostly to a direction in between the northern and southern craters, are thus by around 10°-15° slightly deviated southwards (Fig. 6c). Higher event signal-to-noise ratios, circled in blue, mostly result in BAz between the northern and southern crater region. 6C BAz scatter in a broader range, not focussing on a specific area, but are also rather deviated towards the south in comparison with the reference directions (Fig. 6d). Although some LP events exhibit higher SNRs on the HJZ component, it does not systematically result in back azimuths that better match the INGV-OE reference, suggesting that high SNR alone does not imply more accurate BAz estimates. No clear deviation according to specific sites classified in Fig. 6e is observed either. The array-derived BAz predominantly align approximately 20° south of the reference direction, with deviations reaching up to 40° further south in some cases (Fig. 6f). The 6C BAz measurements also exhibit deviations, with some directions pointing 20° north of the reference and outside the crater region. However, the majority deviate up to 80° to the south relative to the direction indicated by the INGV-OE location (Fig. 6g), resulting in directions far outside the expected range of Mount Etna's crater region. A higher number of high-SNR events is observed at location IV for both methods, which also corresponds to the area with the greatest concentration of located LP events. Whereas INGV-OE and array-derived (Fig. 6c, h) directions are focussed on the crater region, 6C BAz show a broad range reaching outside the crater range (Fig. 6d, h).

### 3.2.5 Slownesses and phase velocities of the LP events

We present the results of phase velocity analyses and spectrograms to furthermore characterize the wave types long-period events consist of. In particular, we aim to determine whether the observed long-period events exhibit characteristics of SH body waves or of Love-waves (surface waves). Calculating phase velocities with the 6C approach for different frequency ranges (Fig. 7a), we observe a decrease of velocity with increasing frequencies. Whereas some LP events yield similar frequencies throughout the event (Fig. 7b), others show an increase of frequencies from the start towards the end of the event (Fig. 7c). The temporal evolution of the phase velocities for all 371 LP events obtained in the frequency range of 0.6 to 1.4 Hz indicates a



**Figure 6.** Overview of all 371 LP event back azimuths and deviation from reference. (a) LP event numbers over time. (b) INGV locations with uncertainty (red) showing LP event counts. (c) BAz for all LP events by the vertical array components. (d) BAz from rotational sensor 6C method. The crater directions are shown as coloured horizontal lines and the legend shows the counts of LP events whose BAz align with the reference (black dots), including uncertainty. (e) numbered INGV locations for (f),(g) plots of BAz deviation to reference versus location. (c),(d),(f),(g) Scatter colour by SNR. (h) Back azimuth histograms.



slight decrease of velocity over the event time (Fig. 7d). Since the event start times were selected manually with a possibility
of timing inaccuracies, the first second is consequently excluded from this analysis.

To enhance the analysis, we also apply array analysis to a selection of volcanic-tectonic (VT) events, where the P- and
S-wave arrival times can be detected precisely. One exemplary VT event with the corresponding BAz and slowness results
is shown in Suppl. Fig. S12. The HHZ components show high semblance for the event start and resulting back azimuth are
in strong agreement with the INGV-OE reference (Suppl. Fig. S12d). Array-derived slowness is at 0.2 s km$^{-1}$ while the p-
wave arrival is visible on the vertical component seismogram (Suppl. Fig. S12a, g). Slownesses observed for s-waves (Suppl.
Fig. S12b) increase non-linearly. A further step-wise slowness increase is observed for the following surface waves. Comparing
these VT event p-, s- and surface wave slownesses with the slownesses obtained for the 371 LP events (Suppl. Fig. S13a), the
array-derived LP event slownesses match expected VT event surface wave velocities from around 0.6 to 1.5 s km$^{-1}$.

## 4   Interpretation and discussion

### 4.1   Limitations and constraints of the applied methods

Array performance depends on the array response defined by the station set-up. Here, Bb17 was placed on a granitic plate with
the rotational sensor, while others were buried directly. Additionally, the event distance should be much larger than the aperture
of the array (Schweitzer et al., 2012). Tremor sources are located 1.2–2.5 km from the array, and LP events at 1.7–2.6 km.
The array aperture is of 0.156 km, thus suitable for analysing events at that distance. Minimum and maximum slowness
(1–2 s km$^{-1}$) and frequency (0.6–1.4 Hz) values of the exemplary LP event (Fig. 5) yield wavelengths of 0.36–1.7 km. At
1.7–2.6 km distance, longer wavelengths may be poorly resolved as the wavefront can no longer be assumed planar (Schweitzer
et al., 2012).

The high uncertainties in the array processing, despite a well-fitting BAz, are likely caused by the array response at the
time of event (Suppl. Fig. S14). The strong main lobe near the slowness grid centre (i.e., at low slowness) leads to multiple
possible BAz solutions within the high-semblance region. A larger array aperture would enhance the BAz resolution for the
lower frequencies (Schweitzer et al., 2012). Based on the general array response Suppl. Fig. S2 and the slowness grid results
for the start of the exemplary LP event (Suppl. Fig. S14), we can conclude, that it primarily emphasizes the main lobe when a
narrow-band coherent signal is present.

Local heterogeneities affect small arrays more than large ones, with near-surface structures distorting the wavefield. For Etna
volcano, these heterogeneities and topographic effects have been used to explain differences in the inversion of slowness data,
with two arrays pointing to different directions of the same recorded tremor signal (Saccorotti et al., 2004), while site effects
do not show a strong influence on the tremor source network locations (Cannata et al., 2013). However, according to Cannata
et al. (2013) and Battaglia et al. (2005), two simultaneously active sources could lead to volcanic tremor sources being located
in between the actual locations by the network amplitude decay technique. LP events and tremor share the 0.6-1.4 Hz range,
complicating separation, but tremor locations from the INGV-OE network match actual volcanic activity well (Cannata et al.,
2013; Sciotto et al., 2022).



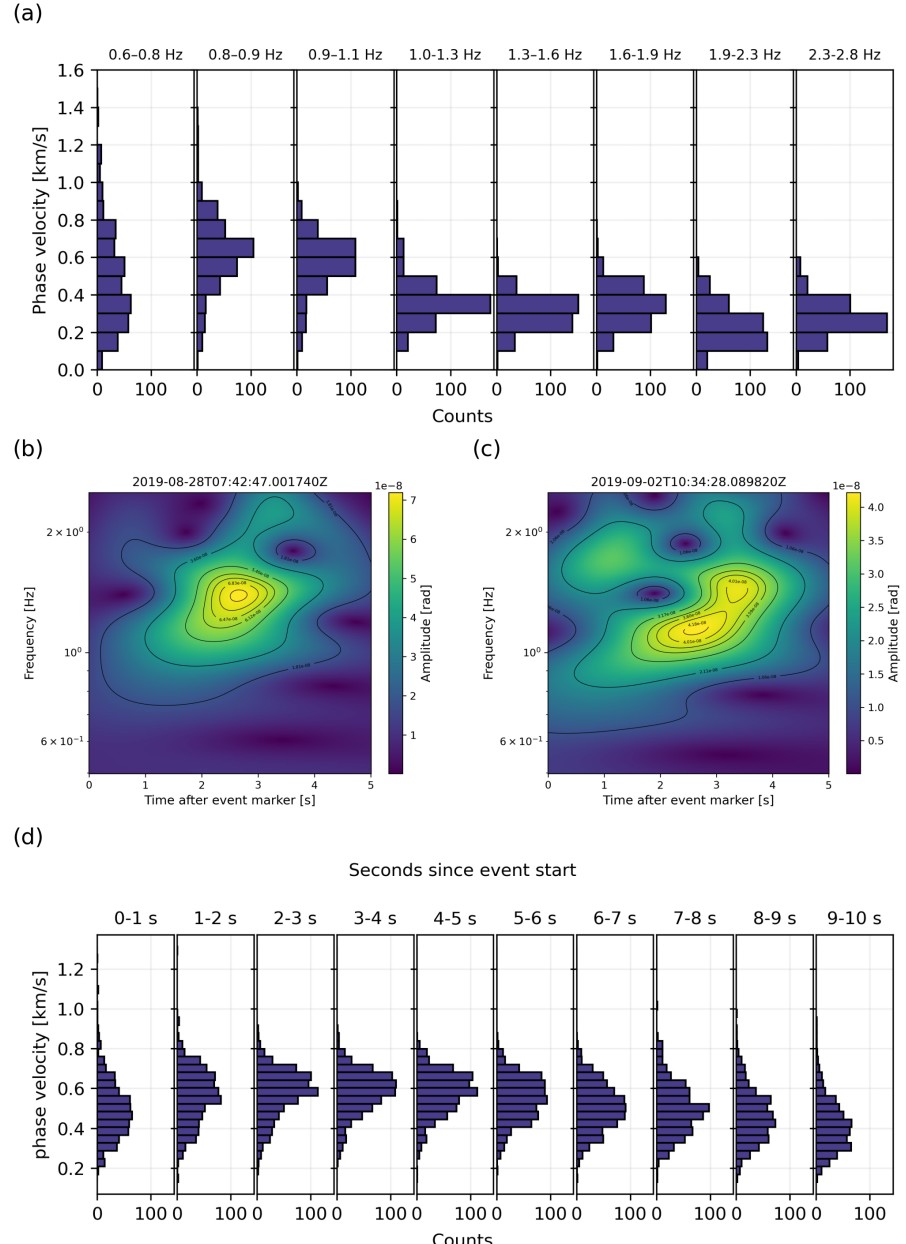

**Figure 7.** Overview of LP event frequency-dependent phase velocity, signal spectrograms, and velocity changes over event time. (a) Phase velocities of the 371 LP events obtained by the 6C method for different frequency ranges respectively. (b, c) Exemplary spectrograms of the high amplitude LP events. (d) Phase velocity evolution over the event time for all the LP events in the 0.6 to 1.4 Hz frequency range.





Finally, we compare the BAz from the array components and the 6C station results to the INGV-OE reference locations assuming that those are close to the true locations. However, the amplitude-based method of the network for tremor is associated with uncertainties of around 500 m. The rotational sensor is sensitive to local heterogeneities which limits the accuracy of
direction results and velocities (Keil et al., 2022; Donner, 2021).

## 4.2 Tremor source parameters: Comparative and joint analysis of the array, 6C, and the INGV-OE network

We plot the INGV-OE tremor locations and observe, despite some outliers, a location clustering according to the eruption phases. Pre-eruptive back azimuths differ between the methods and possibly result from low tremor amplitudes and stronger influence of noise (Fig. 3). From the eruption start with ash emission (phase 1) through Strombolian activity (phase 2) at
the northern craters, INGV-OE and 6C tremor directions point south of the southern craters to SEC and NSEC. Array HHZ directions point to SE and NSE crater. We assume activity at both crater regions during phase 1 and 2, that are respectively indicated by the array directions (southern craters) and visible activity (northern craters). The network and 6C BAz results possibly indicate insufficient high amplitudes to resolve correct directions at the general crater region or might be deviated due to local heterogeneities (Donner (2021) and references therein) that could effectively be mitigated by the array analysis.

During the high amplitude tremor and strong Strombolian activity, INGV-OE and array-derived back azimuths coincide and align with visible activity. They are assumed to be correct. The systematically lower phase 3 back azimuths estimated by the 6C method, compared to those from array and network analyses, raise the question of whether the rotational sensor's 6C method is more sensitive to secondary activity south of the New South East crater, or whether they result from small-scale structures (Keil et al., 2022), given that the 6C method relies on a single-point measurement. Multiple active tremor sources have been
confirmed for an eruption in 1999 on Mt. Etna (Cannata et al., 2008) and can not be excluded for this 2019 eruption. However, we believe that the back azimuth differences by the 6C method are rather caused by local site heterogeneities. We assume that a secondary activity would to some extent as well be displayed by array back azimuths.

The accuracy of array results depends on the semblance of all stations. As semblance is low before any visible activity, the array BAz are not in agreement with the reference (Fig. 4c). At the onset of the eruption, the tremor amplitude increases
slightly with high correlation and semblance for all the methods. Lower array semblance and lower 6C station correlation are observed from phase 3 onwards. At the same time tremor depths changes occur. We hypothesize that the difference in medium propagation paths results in low semblance. While the array slowness reaches similar values during high semblance, the 6C phase velocities represent the tremor amplitude changes and are stable with the exception of phase 3 only, as noted by Eibl et al. (2022a). During phase 3, the reduced 6C station correlation between the seismometer and rotational sensor might be
caused by the deepening of the tremor source and consequently wave type mixing associated with intense Strombolian activity on the surface but tremor in the depth. Eibl et al. (2022a) already noticed SV type of waves during the elevated Strombolian activity. Distance increase by around 5 km and tremor slowness decrease were interpreted as tremor deepening by Eibl et al. (2017a) on Iceland. However, here the distance change is of 0.8 km and might thus not show a significant change in slowness.

We conclude, that low tremor amplitude results in diverging direction estimations by all methods, which could confirm
either noise dominance or secondary sources, as interpreted for LP events by Schick et al. (1982) and tremor by Cannata et al.





(2008) on Mt. Etna. During high tremor amplitudes the phase-based array results fit the INGV-OE reference directions and the visible activity locations, whereas the amplitude-based 6C method might be deviated by local heterogeneities (Keil et al., 2022; Donner, 2021) or due to SV wave types. The tremor slowness decreases with increasing depths while phase velocities increase coherently. Relatively high slowness values of around 1.2 to 1.7 km s$^{-1}$ in comparison to slowness values of 0.5 to 0.7 km s$^{-1}$

for volcanic tremor on Iceland (Eibl et al., 2017a)) could be related to deeper sources, higher complexity of the volcanic edifice and surface wave type tremor. Assuming that array slownesses represent surface waves (1.2-1.4 s km$^{-1}$), we convert them to phase velocities of 0.7 to 0.8 km s$^{-1}$. There is a discrepancy of around 0.3 km s$^{-1}$ compared with the phase velocities derived using the 6C method. This might be, as before mentioned, due to small-scale structures deviating the incoming waves (Keil et al., 2022; Donner, 2021), or due to a mixtures of wave types during phase 3 (Eibl et al., 2022a), resulting in slower phase

velocities.

### 4.3 Love wave type LP events: velocities and directions

Even though the eruption, starting on 4 September 2019 with weak ash emissions, took place at the North East and Voragine craters (Eibl et al., 2022a), most LP events point to a source at shallow depth further north of the North East crater according to the network directions. LP events at Mt. Etna have in other studies already been mentioned to be present without volcanic

activity as well, thus not directly related to volcanic activity (Cannata et al., 2009).

     For the exemplary LP event on 2 September 2019 no clear direction change is observed comparing the BAz of the small-amplitude signals emerging around 15:50:45 and 15:51:05 in Fig. 5a, b, - prior to the detected LP event - to the LP event (Fig. 5). Persistent low amplitude activity could explain similar directions before and after the defined LP event.

     We achieve better fitting BAz from the array processing for all the LP events than from the 6C analysis. However, both

methods seem to be influenced by a deviation towards south, - the array mostly by up to 20° and the rotational sensor method by even up to 80°, but generally more scattered results in comparison with the INGV-OE directions. Disagreement could possibly be related to the faults indicated in Fig. 1d, which have already been found to cause short wavelength discrepancies in the comparison of strain recorded by Distributed Dynamic Strain Sensing (DDSS) and the array derived strain (Currenti et al., 2021; Jousset et al., 2025). Station *Bb22*, which was also used in this study, seemed to be most affected by the faults (Currenti

et al., 2021). However, no significant waveform differences were observed between the array stations. The faults may affect the entire array, not just the nearest station. The INGV-OE network locations could furthermore be affected by up to 500 m uncertainty, making it difficult to even distinguish between northern and southern craters. If this separation is not possibly due to the uncertainties, then array back azimuths point at least mostly to the summit region, fitting visible activity generally. 6C BAz results, from a point measurement, are possibly again affected by local heterogeneities, as LP event amplitudes and LP

event SNR for the HJZ component (Fig.6d) did not show a clear effect on the BAz reference agreement. Low 6C SNR relative to the array, however, likely contributes to the increased scatter observed in the results (Brotzer et al., 2025).

     The frequency increase over the LP event time is interpreted as dispersion. Together with the 6C phase velocity decrease with increasing frequencies surface waves are expected (Udías and Buforn, 2018; Bormann et al., 2012; Chouet et al., 1997). Rotational sensor recordings suggest a SH-wave dominated wavefield and the array-derived slownesses are typical for surface



waves. The joint interpretation of array and rotational sensor results hence allow us to constrain that long-period events at Etna between 24 August to 6 September 2019 were predominantly composed of Love-waves. 6C LP phase velocities of mostly 400-700 ms$^{-1}$, as mentioned by Eibl et al. (2022a) match the shear wave velocity range of 400-1100 ms$^{-1}$ obtained from distributed acoustic sensing (DAS) investigation on Mt. Etna (Jousset et al., 2022). Zuccarello et al. (2016) obtained the same range of shear wave velocities (400-700 ms$^{-1}$) down to 130 m depth. Tremor array slowness values are around 1.2 s km$^{-1}$

with high semblance and agree with expected surface wave slownesses from VT event analysis. The 6C phase velocities are related to higher slowness values of 2.2 to 2.8 s km$^{-1}$ which is possibly a cause of local scattering.

## 5    Conclusion and Outlook

For the first time in a volcanic setting, we present a direct comparison between back azimuths derived from a seismic array and those obtained from a six-component-station of a rotational sensor and a seismometer. Differences in tremor and long-period

event back azimuths between the two methods and in comparison with the network reference of the INGV-OE highlight the complexity and influence of the volcanic edifice. We conclude that low amplitude tremor leads to back azimuth disagreement between all three methods and none leads to directions indicating the actual surface activity. During high amplitude tremor, the INGV-OE network and array BAz overlap and point to the northern craters where strong Strombolian activity was observed. The 6C BAz however are by 20° deviated south, which is interpreted to result from topography or local heterogeneities. The

array tremor slownesses indicate reliable results with slightly faster velocities during the eruption confirming a deeper tremor source. 6C phase velocities are expected to be higher and the disagreement might derive likewise from topographic influence or local structures. Eibl et al. (2022a) noted a mixed tremor wavefield during strong Strombolian activity. As the 6C method relies on the SH-type waves only, it can not be expected to represent the mixed wavefield directions accurately. Applying another method for SV-type waves on the horizontal components of the rotational sensor (Wassermann et al., 2020) resulted

in expected directions towards the northern craters (Eibl et al., 2022a). Likewise, applying array processing on the horizontal array components for the low amplitude eruption phases mainly consisting of SH-type waves should result in more accurate directions too. We thus suggest adapting the methodology according to the wavefield once known.

     As LP event back azimuths show southward deviation from the INGV-OE reference for both methods, we assume a strong topographic effect due to the volcanic site itself (Schick et al., 1982), which could to some extent also affect the array (Eibl

et al., 2017a; Chouet et al., 1997). Nonetheless, array-derived BAz encompass the entire crater region and network locations yield uncertainties comprising the entire summit area as well. Scattered 6C BAz could derive from local heterogeneities (Singh et al., 2019) or the distance of the 6C set-up to the events (Izgi et al., 2025), as a direct influence of event SNR and amplitude could be excluded.

     Sciotto et al. (2022) stated, that tremor amplitudes at Mt. Etna are more connected to volcanic activity, than the LP event

amplitudes, which could be confirmed in terms of locations in our work. They deduce from their findings, that tremor and LP events at Mt. Etna most likely have different source mechanisms. Further possible interpretation of no clear correlation between LP events and eruption are given by De Barros et al. (2009); Cannata et al. (2009); Sciotto et al. (2022).

Adding more stations improves the array's ability to filter seismic signals based on their slowness and direction, allowing for better isolation of the target signals and suppression of unwanted energy (Schweitzer et al., 2012). Even more information could be obtained from a second array together with another 6C station at two different locations to derive event locations instead of directions while compensating for structural effects on the wavefield (Métaxian et al., 2002). Relocating the instruments to a different site of Mount Etna could offer a more detailed understanding of the path effects mentioned by Saccorotti et al. (2004). Finally, we recommend further investigation of the low-amplitude signals preceding the network-detected LP events to better constrain source information. Fibre optic technologies, of which rotational sensors, are promising for volcano monitoring in terms of understanding volcanic processes and structure (Jousset et al., 2025). We recommend more testing of rotational sensors in volcanic environments to understand specific influence of topography and scattering on both methods. While the array processing appears to provide more accurate directions of tremor and LP event signals, the rotational sensor facilitates wavefield separation. Our study's outcomes highlight the advantage of combining both approaches, resulting in improved volcano-seismic source characterization.

*Data availability.* The seismic data will be made publicly available at GEOFON during the review process. Preprocessed data files will be made available through GFZ Data Services. The rotational sensor and Bb17 data is available at GEOFON Eibl et al. (2022b) via https://doi.org/10.14470/ME7564062119.

*Author contributions.* PJ, EE, and GC planned the campaign; EE, GC, and PJ installed the sensors; EE developed the conceptualization; NV and EE analysed the data; NV developed the visualisations and wrote the manuscript draft; EE, MO, GC, GdG, MS and PJ reviewed and edited the manuscript.

*Competing interests.* At least one of the (co-)authors is a member of the editorial board of Solid Earth.

*Acknowledgements.* We thank the "Geophysical Instrument Pool Potsdam" from GFZ for providing and the "Parco dell'Etna" and the municipalities (Linguaglossa and Castiglione di Sicilia) for the permit to deploy the instruments in the Etna national park. This work was possible thanks to support with logistics and in the field of Daniel Vollmer from University of Potsdam and Danilo Contrafatto, Graziano Larocca, Daniele Pellegrino and Mario Pulvirenti from the Istituto Nazionale di Geofisica e Vulcanologia-Osservatorio Etneo (INGV-OE). Many thanks go to Gizem Izgi for improving comments and discussion.

*Financial support.* This research has been funded by DFG. Grant number EI 1099/4-1.



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
