# Peer review of "Constraining the wavefield of volcano-seismic events on Mt. Etna, Italy through rotational sensor and array observations"

_EGUsphere, 2025_

## Author Comment (AC1)

*We thank the reviewer for their considered and insightful review that helped improving our manuscript.*

*This manuscript presents a detailed analysis of the wavefield excited by tremor and long-period (LP) events associated with the eruptions of Mount Etna, Italy, during August–September 2019. In particular, the authors use an array composed of a rotational sensor and six seismic stations to estimate the back azimuth and phase velocity of seismic waves generated by these events. By comparing the array-derived back azimuths with the source locations of tremor and LP events estimated from INGV-OE routine processing, they demonstrate a high level of agreement during periods of intense tremor activity, whereas discrepancies are observed in the rotational sensor data. The authors attribute these inconsistencies to local structural heterogeneities. Furthermore, combined analysis of the rotational sensor and seismometers reveals that SH waves are dominant in the wavefield of both tremor and LP events. The integration of rotational sensor data with seismic array observations is highly innovative and provides valuable insights for future observations using rotational sensors. Below are several comments that may help improve the manuscript.*

Length of the Time Window for Tremor Analysis

In this study, a relatively long time window of 30 minutes was used for tremor analysis. However, the waveform characteristics may vary during such a long period. In cases of non-stationary seismic activity or changes in propagation paths, averaging over this interval may obscure temporal variations. Including supplementary analyses to evaluate the stability of the waveforms within each window would enhance the reliability of the results.

*XX We tested different sliding window lengths of 0.5 and 5 minutes for the tremor analysis. We now added a supplementary figure (S3) that shows the comparison of back azimuth and slowness from those time windows. The results of our manuscript are not affected by the window length as trends in changes of back azimuth and slowness remain consistent regardless of the sliding window duration.*

Significance of Back-Azimuth Variations

During phases 0–1, the back azimuth is reported to change from 210° to 190°, but the estimation uncertainty is ±10–20 degrees. Considering this margin of error, a change of about 20 degrees may not be statistically significant, and the interpretation based on this variation should be made with caution. In addition, the authors note that the back-azimuth estimates derived from different methods (array, 6C method, and INGV network) show different directions across phases 0–2. However, if all values fall within their uncertainty ranges, emphasizing inter-method differences may not be meaningful. Please clarify whether these differences are statistically significant, or at least interpret the results with due consideration of the uncertainties.

*XX The use of denser real-data sampling (0.5-minute sliding windows for tremor and higher overlap for the LP event sliding time windows) enables us to compute standard-deviation-based uncertainties, improving robustness and avoiding the elevated uncertainties due to the long time windows before. These new uncertainties are more accurate, but still reach around 10°. We therefore adapted the text accordingly and thank the reviewer for the comment improving this section.*

In Figure 2, the plotted colors for phases P2 and P4 are quite similar, making them difficult to distinguish, especially in grayscale printing or for readers with color vision deficiencies. Consider using more distinct hues (e.g., blue and red) to improve visual clarity.

*XX We modified the colors of phases P2 and P4 to yellow and orange, making them separable as well in black and white while still showing a certain similarity between those phases. Vertical bars in b)-d) were added to improve visibility too.*

In Figures 4c (array-derived slowness) and 4d (6C-derived phase velocity), uncertainties or confidence intervals of the estimated values are not indicated, making it difficult to assess the reliability of the results. When comparing outcomes obtained from different methods, it is essential to evaluate and display these uncertainties. If possible, please include error bars.

*XX We now added uncertainty bars for the array derived slowness and 6C phase velocities obtained by the standard deviation of smaller time window results, as done for the back azimuth.*

In Figure 5d, high correlation coefficients are observed not only during LP-event periods but also at other times. It is unclear whether these correlations correspond to real events or to noise signals. Please provide a clear explanation in the text regarding the possible causes of these high correlations.

*XX We added more explanation in the interpretation section (5.3). It is likely due to LP event pulses with lower magnitude, from the same direction, which were not picked as LP events in the INGV-OE catalogue.*

Figure 6 contains a large amount of diverse information (temporal changes, spatial distribution, directional deviations, histograms, etc.) within a single figure, making it difficult for readers to follow. The following reorganization is suggested:

Arrange panels (a), (c), and (d) vertically to align their time axes and clarify temporal consistency.

Combine panels (f), (g), and (h), which contain statistical information on back-azimuth deviations, into a separate figure.

Enlarge and reposition the maps (b) and (e) for improved readability.

*XX We split Figure 6 into two figures, re-arranging the plots and aligning shared axes. Map plots are now larger.*

In Figures 7b and 7c, it is not specified which instruments (array or rotational sensor) and which components (e.g., HHZ, HJZ) the running spectrograms are based on. Please indicate this information clearly in the figure captions or in the main text.

*XX We clarify that we used the rotational sensor HJZ component to create the spectrograms.*

---

## Author Comment (AC2)

*We thank the reviewer for their detailed review to improve the manuscript and text flow.*

*Summary*

*The authors compare the estimations of backazimuths and phase velocities from array data as well as a rotational sensor (6C) with reference results from the permanent INGV network deriving implications for the dominant wavetype of the recorded LP- and tremor events. Due to the significant complexity of the involved wavefields the authors point out that conventional array recordings often do not allow for a detailed wavefield composition regarding tremor and LP events and this lack of information forms the motivation for this study introducing rotational sensor data.*

*Reference localisation of tremor and LP-events (INGV) are explained in great detail as well as the derived backazimuth values that are ultimately used for comparison. High-quality figures contain a very high amount of information in very good detail. The authors reveal an interesting discrepancy between backazimuths derived from rotational sensor data compared to traditional array or network methods. The systematic offset observed for the rotational sensor is likely attributed to site effects which are substantial in the heavily scattering medium on Mt Etna's edifice to which the single rotational sensor would be particularly susceptible. The possibility of wave type mixing is pointed out due to deeper tremor sources and simultaneous surface strombolian activity. Love-wave dominated wave fields are expected for the LP events according to the rotational sensor data. The authors conclude that while the single rotational sensor can not provide reliable information in terms of direction of arrival of tremor or LP signals it is a useful addition to traditional array or network-based analysis as it offers wave field separation. Multiple arrays would improve results significantly as source locations rather than direction of arrival may be obtained while also multiple rotational sensors would reduce effect of heterogeneities.*

*Some comments/suggestions below*

*general comments on sections or figures*

section 2.2 --- how do different window lengths for calculation of RMS of tremor affect results? The tremor sliding window is relatively long, potentially obscuring shorter time variations. A shorter time window might be an interesting test or alternatively a larger overlap of adjacent windows for better time resolution. As for LP events, would a different 5 second noise window (say 15 seconds before event window instead of 25) affect SNR in a meaningful way or would the distribution of SNR across all events remain roughly the same?

*XX We tested two smaller sliding time windows which are presented in Figure S3. The results of our manuscript are not affected by the window length, smaller time windows lead to a broader range of back azimuth and slowness, but they show similar trends as the 30-min window results. We thus don't expect shorter time variations. We tested noise windows for the SNR calculation defined 10 s prior to the event time and compared those to the results of 30 s prior the event time. The observed SNR differences show no systematic offset, with positive and negative deviations distributed randomly and canceling out on average. Based on these findings, we chose to stick with the 30 s noise window, as for about one-third of all LP events, smaller amplitude peaks were observed around 5 s prior to the event start time (Section 4.2.3)*

section 2.5 --- the detailed explanation of the array processing may be condensed a little more. Array processing is quite routine and since a reference to more detail is given anyway (line 207) this section could be kept shorter.

*XX We understand the remark. For the purpose of methodological comparison, we do however want to include a brief description of the array processing too. Including equations requires describing parameters, which limits the possibility of shortening the text.*

Section 3 --- the estimated uncertainty of derived backazimuths is on the same order as reported variations between the 3 methods or some of their changes over time, therefore observed temporal variations or between methods need to be checked in terms of statistical relevance as they may not be significant

*XX We modified the text according to the (new enhanced) uncertainties. During certain times variations are indeed not significant considering the uncertainties. We thank the reviewer for this remark.*

Figure 1 --- include reference to fault locations in figure caption

*XX We added the reference to the fault locations in the caption of the figure.*

Figure 2 --- colours for tremor locations of P2 and P4 are too similar | in the caption the array center is referred to as "yellow dot" when it is a green square according to legend. "Array centre" could be removed entirely as it covers the symbol for the rotational sensor in plot

*XX The colours of P2 and P4 have been modified accordingly. The array centre, as suggested, is not displayed in the figure anymore.*

Figure 3 --- move legend in panels b, d to respective top left corners to avoid overlap with data | in panels b-d uncertainty bars could be added to legend as well for quicker overview

*XX Legends of Figure 3b and 3d have been moved, - the uncertainty bars have been added to the legends too.*

Figure 4 --- slowness in panel c could be colour-coded according to event altitude as well to better track corresponding data points in panels b and c | in caption remove "The" in last sentence

*XX We thank the reviewer for this suggestion. The slowness displayed in Figure 4c is now colour-coded, but according to hypocentral distance, as the event altitude can easierly be compared by colours of b) as x-axes are shared. We do not notice a unique change of slowness with decreasing distance. "The" was removed in the last sentence of the caption.*

Figure 6 --- if the subplots could be re-arranged in such a way that the entire figure does not have to be tilted but fits onto the page normally this would improve readability

*XX The entire Figure 6 has been split up in two as suggested by Reviewer 1.*

section 3 --- the listing of all results could be condensed a little to increase overall flow or reading as some parts are a little drawn-out

*XX We shortened this section where possible.*

for all figures (including supplementary material) fontsizes of axes or colorbar labels, ticks and legends could be increased a bit for better readability

*XX We enlarged fontsizes where it was possible.*

suggestions regarding text flow

lines 74-77 --- The aim of this study is to test, for the first time in a volcanic environment, whether the 6C approach provides reliable estimates of back azimuths and velocities compared with those obtained from a conventional seismic array, and whether these results are consistent with reference locations from the Istituto Nazionale di Geofisica e Vulcanologia–Osservatorio Etneo (INGV-OE) network.

line 102 --- Using a rotational sensor Eibl et al. (2022a) have shown, that

lines 114-115 --- For each LP event, we calculated the RMS within a 5-second time window containing the signal as well as the RMS within a 5-second noise window that

line 249 --- just use NEC instead of North East crater as acronym was introduced earlier

line 290 --- 1.6 km altitude, a trend which is statistically significant despite the uncertainties.

line 308 --- remove "the"

line 321 --- the BAz is consistent with

line 348-349 --- to a direction in between the northern and southern craters about 10°-15° further south compared to the INGV reference

line 351 --- but also deviate

line 363 --- remove "furthermore"

line 433 --- However, in this case the distance changes by 0.8 km

line 449 --- In previous studies, LP events at Mt. Etna have been mentioned

line 476 --- which is possibly related to local scattering

line 484 --- The 6C Baz, however, point back in a direction 20° further south, which

*XX We modified all suggestions accordingly and thank the reviewer for enhancing the text flow.*